# Integrated Transcriptome and Metabolome Analyses Reveal Details of the Molecular Regulation of Resistance to Stem Nematode in Sweet Potato

**DOI:** 10.3390/plants12102052

**Published:** 2023-05-22

**Authors:** Shouchen Qiao, Jukui Ma, Yannan Wang, Jingwei Chen, Zhihe Kang, Qianqian Bian, Jinjin Chen, Yumeng Yin, Guozheng Cao, Guorui Zhao, Guohong Yang, Houjun Sun, Yufeng Yang

**Affiliations:** 1Cereal Crop Research Institute, Henan Academy of Agricultural Sciences, Zhengzhou 450002, China; 13676963200@163.com (S.Q.); alman001@hnagri.org.cn (Y.W.); kangzhihe@163.com (Z.K.); bianqianqian11@163.com (Q.B.); yinyumeng0521@163.com (Y.Y.); yangguohong9211@163.com (G.Y.); 2Xuzhou Institute of Agricultural Sciences in Jiangsu Xuhuai Area, Xuzhou 221000, China; majukui@126.com (J.M.); ibcjw0825@126.com (J.C.)

**Keywords:** sweet potato, stem nematode, transcriptome, metabolome, ethylene, secondary metabolites, resistance pathway

## Abstract

Stem nematode disease can seriously reduce the yield of sweet potato (*Ipomoea batatas* (L.) Lam). To explore resistance mechanism to stem nematode in sweet potato, transcriptomes and metabolomes were sequenced and compared between two sweet potato cultivars, the resistant Zhenghong 22 and susceptible Longshu 9, at different times after stem nematode infection. In the transcriptional regulatory pathway, mitogen-activated protein kinase signaling was initiated in Zhenghong 22 at the early stage of infection to activate genes related to ethylene production. Stem nematode infection in Zhenghong 22 also triggered fatty acid metabolism and the activity of respiratory burst oxidase in the metabolic pathway, which further stimulated the glycolytic and shikimic pathways to provide raw materials for secondary metabolite biosynthesis. An integrated analysis of the secondary metabolic regulation pathway in the resistant cultivar Zhenghong 22 revealed the accumulation of tryptophan, phenylalanine, and tyrosine, leading to increased biosynthesis of phenylpropanoids and salicylic acid and enhanced activity of the alkaloid pathway. Stem nematode infection also activated the biosynthesis of terpenoids, abscisic acid, zeatin, indole, and brassinosteroid, resulting in improved resistance to stem nematode. Finally, analyses of the resistance regulation pathway and a weighted gene co-expression network analysis highlighted the importance of the genes *itf14g17940* and *itf12g18840*, encoding a leucine-rich receptor-like protein and 1-aminocyclopropane-1-carboxylate synthase, respectively. These are candidate target genes for increasing the strength of the defense response. These results provide new ideas and a theoretical basis for understanding the mechanism of resistance to stem nematode in sweet potato.

## 1. Introduction

Sweet potato (*Ipomoea batatas* (L.) Lam) is the fifth most important food crop in the world [1]. As well as being an important food and feed crop, it is a useful industrial raw material and new energy crop in China. As such, it plays an important role in food and energy security. Therefore, there is an increasing amount of research on sweet potato worldwide.

Stem nematode (*Ditylenchus destructor* Thorne, SN) is a serious disease that limits the production of many crops such as sweet potato, potato, peanut, and onion. This disease usually reduces sweet potato yield by 20% to 50%, but serious infections can destroy the entire crop [2]. The breeding of SN-resistant cultivars has become one of the most important objectives in sweet potato breeding programs, and many studies have focused on functionally characterizing genes related to SN resistance.

The sweet potato gene *IbMIPS*, which encodes myo-inositol-1-phosphate synthase, is homologous to a SN-resistance gene in sugar beet. Transgenic sweet potato plants overexpressing *IbMIPS1* showed increased SN resistance as well as tolerance to abiotic stress [3,4,5]. Qu et al. [6]. acquired several nucleotide-binding–leucine-rich repeat (NBS–LRR) homologous sequences showing close relationships with SN resistance from the sweet potato SN-resistant cultivar AB94078-1, the susceptible cultivar Xushu 18, and their eight progenies. Genes encoding nucleotide-binding site (NBS) proteins are a major type of resistance (R) gene that play important roles in biotic stress responses in plants. Si et al. [7,8]. recently analyzed the genome and transcriptome data of *Ipomoea* species and isolated several NBS genes responding to SN infection. These genes have potential value for improving the SN resistance of sweet potato. A suppression subtractive hybridization analysis revealed that four genes related to SN resistance and one gene encoding a serine/threonine kinase protein showed different expression patterns in the SN-resistant cultivar Lushu 3 [9]. Overexpression of *IbMVD*, encoding mevalonate diphosphate decarboxylase, increased the SN resistance of transgenic sweet potato [10]. Similarly, overexpression of *OCI* encoding the protease inhibitor cystatin-I from rice significantly increased the SN resistance of transgenic plants [11]. Another study aimed to develop molecular markers related to SN resistance and obtained the SN resistance-related marker 0PD689 using PCR amplification of the F_1_ population of the resistant parent Xu 781 and the highly susceptible parent Xushu 18 [12]. Zhao et al. [13] developed four sequence-related amplified polymorphism markers closely related to SN-resistance genes of sweet potato, and they have proved to be a useful reference for gene mapping.

To date, most studies on SN resistance in sweet potato have focused on the functional characterization of single genes. However, it would be useful to gain an overview of SN-responsive pathways in sweet potato. In this study, therefore, we compared the transcriptomes and metabolomes between the SN-resistant cultivar Zhenghong 22 and the susceptible cultivar Longshu 9 after SN infection. The resulting profile of SN-defensive pathways provides a valuable reference for understanding the SN-resistance mechanism of sweet potato.

## 2. Results

### 2.1. Phenotypes of Resistant and Susceptible Cultivars after Infection with Stem Nematode

In the early stage of infection (0–10 d), there was no significant difference in the phenotype of storage roots between Zhenghong 22 and Longshu 9, but at 10 dpi, the stem nematodes had spread into the flesh in the susceptible cultivar Longshu 9. Almost the entire storage root of Longshu 9 was infected at 30 dpi, and its degree of infection was significantly higher than that in storage roots of Zhenghong 22 (Figure 1).

### 2.2. Transcriptomic Differences between Resistant and Susceptible Cultivars

The transcriptomes of resistant and susceptible cultivars were compared to identify genes related to SN resistance and the corresponding pathways at different stages after infection. In total, 49,283 genes were detected from the transcriptome sequences. At 0.5 dpi, 2667 genes were upregulated and 2681 genes were downregulated in the resistant cultivar compared with the susceptible cultivar. At 1 dpi, 2319 genes were upregulated and 2936 genes were downregulated in the resistant cultivar compared with the susceptible cultivar. At 3 dpi, 2199 genes were upregulated and 2274 genes were downregulated in the resistant cultivar compared with the susceptible cultivar. At 10 dpi, 2978 genes were upregulated and 4648 genes were downregulated in the resistant cultivar compared with the susceptible cultivar. At 30 dpi, 3047 genes were upregulated and 3900 genes were downregulated in the resistant cultivar compared with the susceptible cultivar. As illustrated in Figure 2A and Appendix A, the comparative transcriptome analysis revealed differences between the two cultivars after infection, especially at 0.5 dpi, 10 dpi, and 30 dpi. The Venn diagram in Figure 2B shows that the differences in transcriptomes between the resistant and susceptible cultivars were smallest at 1 dpi and 3 dpi, and largest at 10 dpi, followed by 30 dpi, and then 0.5 dpi (Figure 2B). The same results are shown in the principal component analysis (PCA) plot (Figure 2C).

After removing the genes that were co-expressed in the control (see Venn diagram), the remaining genes expressed in Zhenghong 22 during SN infection were subjected to Gene Ontology (GO) analysis. In the biological process category, the GO terms enriched with DEGs were involved in protein phosphorylation and phosphorylation pathways, consistent with the increased activity of signal transduction pathways to mediate disease resistance (Figure 3A). The third pathway was the defense response, indicating that Zhenghong 22 initiated a stronger defense response than Longshu 9. The salicylic acid (SA) pathway and the ethylene pathway were also activated in Zhenghong 22.

After screening 5709 DEGs, there were 1277 genes with pathway annotations in the Kyoto Encyclopedia of Genes and Genomes (KEGG) (Figure 3B). The pathways significantly enriched with DEGs were in the metabolism, genetic information processing, environmental information processing, cellular processes, and organismal systems categories. Among them, the metabolism and environmental information processing pathways were strongly related to SN resistance. Some DEGs related to the synthesis and metabolism of amino acids were upregulated in the resistant cultivar Zhenghong 22, consistent with the metabolomics data. Other upregulated pathways in Zhenghong 22 included the N-glycan synthesis pathway; the tropane, piperidine, and pyridine alkaloid biosynthesis pathway; carotenoid biosynthesis, terpenoid backbone biosynthesis, other secondary metabolic pathways, the mitogen-activated protein kinase (MAPK) signaling pathway; the ethylene and abscisic acid (ABA) pathway; the leucine-rich repeat (LRR) signaling pathway; the brassinosteroid-insensitive associated kinase receptor (BAK) signaling pathway; and the plant-pathogen interaction pathway. In addition, *WRKY* superfamily genes were upregulated in Zhenghong 22. The transcript levels of most genes in the ethylene pathway were higher in Zhenghong 22 than in Longshu 9. The calcium ion transport pathway mediating peroxide accumulation was enriched with DEGs in Zhenghong 22. These results were consistent with those obtained in the trend analysis and the analysis of DEGs and differentially accumulated metabolites (DMs) showing at least 10 times difference in abundance between pairs of samples (Appendix A).

### 2.3. Comparative Metabolomic Analysis between Resistant and Susceptible Cultivars

On the basis of phenotypic changes after infection, samples at 0 dpi, 10 dpi, and 30 dpi were collected for metabolomic analysis. The sample cluster analysis distinguished all samples of the resistant cultivar Zhenghong 22 from those of the susceptible cultivar Longshu 9 (Figure 4A). The data quality and trends detected in the positive and negative ion modes were basically the same (Appendix A), and the data quality was good. There were greater differences in metabolites between the control and 30 dpi samples than between the control and 10 dpi samples (Figure 4B).

Analyses of the metabolomics data revealed 16,916 metabolites, of which 7849 were detected in the positive ion mode (POS). Of them, 1094 metabolites were annotated, and 6755 metabolites were unknown. In total, 9067 metabolites were detected in the negative ion mode (NEG). Of them, 610 metabolites were annotated, and 8457 metabolites were unknown. Overall, 423 DMs were detected between the resistant and susceptible cultivars at 10 dpi and 30 dpi (Figure 5A). The KEGG enrichment analysis showed that the DMs were mainly in metabolic pathways (Figure 5B). Amino acid metabolism and the contents of various amino acids were increased in Zhenghong 22, as was the biosynthesis of plant secondary metabolites and the contents of metabolites involved in the citrate cycle.

In Zhenghong 22, the DMs included many in the ABA synthesis pathway, ethylene synthesis pathway, and SA synthesis pathway. The increased content of the ethylene precursor 1-aminocyclopropane-1-carboxylate (ACC) suggested that this pathway was stimulated to produce more ethylene to participate in SN resistance. The SA content increased sharply after infection in both cultivars, indicating that they both initiated the SA pathway in response to SN infection, but the content of the SA precursor trans-cinnamic acid was higher in Zhenghong 22 than in Longshu 9. The results of the trend analysis and the adjusted differential factor analysis were consistent with these findings (Appendix A).

### 2.4. Screening of Candidate Genes and Metabolites

To process the large amount of transcriptome data, we conducted a WGCNA to obtain information about clusters of co-expressed genes. The Midnight Blue and Green modules were related to Zhenghong 22 at all times after infection. The Darkseagreen4, Orangered3, Skyblue3, and Coral2 modules were related to Zhenghong 22 at 0.5 dpi. Coral2 was related to both 0.5 dpi and 30 dpi, while Skyblue3 was only related to 30 dpi (Appendix A). The 37 genes (Table 1) with the highest connectivity with the modules were screened, and the GO enrichment analyses included a total of 28 key genes that were enriched in 35 GO terms. The genes *itf15g12000.t1* and *itf15g12000.t2* were enriched in the GO terms “responses to insects” and “stimulus response function”. The genes *itf03g17570.t1* (*XI-K*), *itf04g13120.t1*, *itf05g19720.t1*, *itf07g02790.t1*, *itf10g23050.t1*, *itf13g03200.t1* were also enriched in the “stimulus response function” GO term. Only nine key genes had pathway annotations; *itf06g12490.t1* belonged to the LRR receptor protein kinase family and was enriched in the MAPK signaling pathway, which is also considered as a plant–pathogen interaction pathway. *Itf12g04740.t1* was enriched in the plant hormone signal transduction pathway. *itf10g2060.t1* encoding peroxidase was enriched in the phenylpropanoid biosynthesis pathway. *itf15g12000.t1* and *itf15g12000.t2* were enriched in the fatty acid metabolism pathway, and other key genes were enriched in the carbon metabolism, purine metabolism, and nucleotide repair pathways.

To identify candidate metabolites related to infection, we focused on metabolites that accumulated after infection, as they show strong positive relationships with the disease resistance response. After removing basic metabolites such as amino acids, the remaining 19 were identified as candidate metabolites. The WGCNA analysis revealed Zhenghong 22-related modules and metabolites with higher connectivity in these modules. For most of the metabolites with higher connectivity, their chemical formulae and related pathways were unknown. Of the 13 metabolites with molecular formulae, 11 showed more than 10 times difference in abundance between the resistant and susceptible cultivars and were selected for further analysis (Table 2).

### 2.5. Combined Transcriptome and Metabolome Analysis

The transcriptome analysis revealed details of gene expression during the defense response after infection, and but only a few metabolites had molecular formula and pathway annotation information available. The top 250 DEGs and DMs with correlation coefficients were analyzed, and the top 100 were selected for heatmap analysis. As shown in the heatmap, there were strong correlations between the DEGs and DMs (Appendix A).

We determined which pathways were enriched with DMs and DEGs at 0, 10, and 30 dpi. The top five pathways showing differences between the two cultivars before infection were starch metabolism, sucrose metabolism, metabolic pathway, tyrosine metabolism, and carotenoid biosynthesis. At 10 dpi, the top five pathways showing differences between the two cultivars were metabolic pathway, biosynthesis of secondary metabolites, biosynthesis of monoterpenes, metabolism of nicotine and nicotinamide, and biosynthesis of benzoquinone; these findings were consistent with the results of the transcriptome and metabolome analyses. The phenotypes of the two cultivars were not very different at 10 dpi, but the defense response had clearly been initiated in Zhenghong 22. There were significant differences in the phenotype between the two cultivars at 30 dpi, and larger differences in pathways enriched with DEGs and DMs. The pathways enriched with DEGs and DMs at 30 dpi included those detected at 10 dpi as well as glutathione metabolism, carotenoid biosynthesis, glycolysis, fatty acid degradation, linoleic acid metabolism, and the pentose phosphate pathway.

The criterion for DEGs and DMs was increased to at least 10 times difference between the two cultivars, and the upregulated genes and metabolites in Zhenghong 22 were subjected to KEGG analysis. The following nine pathways were enriched with strongly up-regulated DEGs in Zhenghong 22 after infection: Biosynthesis of secondary metabolites; phenylpropanoid biosynthesis; tropane, piperidine, and pyridine alkaloid biosynthesis; fatty acid degradation; monoterpenoid biosynthesis; nicotinate and nicotinamide metabolism; MAPK signaling pathway-plant; and terpenoid backbone biosynthesis. Four pathways were enriched with strongly upregulated DMs in Zhenghong 22 after infection: biosynthesis of alkaloids derived from the shikimate pathway; the citrate cycle (TCA cycle); biosynthesis of phenylpropanoids; and biosynthesis of plant hormones.

A correlation network diagram was constructed to discover relationships between candidate metabolites and candidate genes. We selected 35 genes with high connectivity out of 45 candidate genes and 27 out of 32 metabolites. The candidate genes and metabolites showed high connectivity. The correlation network diagram divided them into two groups (Figure 6 and Appendix A). The first group (on the left in Figure 6) consisted of two genes in the stimulus response pathway, an unknown gene, a gene encoding a DNA-directed DNA polymerase, and a gene encoding a Ser/Thr protein kinase (*Itf14g18430.t1*). The *Itf14g18430.t1* gene triggered the initiation of the insect resistance defense, and it connected the two modules. The metabolites in this module included amino acids, peptides, linoleic acid, and metabolites of the TCA cycle. These metabolites are related to the accumulation of raw materials and energy in the early stage of infection. The other module (Figure 6, on the right) contained genes encoding disease resistance proteins, genes involved in the stimulation response, and genes encoding subtilisins, chitinases, and receptor kinases. The metabolites in this module included those related to the secondary metabolite synthesis pathway, hormone synthesis pathway, benzoin synthesis pathway, terpenoid backbone formation pathway, and alkaloid synthesis pathway. On the basis of these findings, six genes related to disease resistance were selected for qRT-PCR verification (Figure 7A–F).

### 2.6. Analysis of Regulatory Pathway Involved in the Response to Stem Nematode in Sweet Potato

The genes and metabolites that were upregulated in the resistant cultivar Zhenghong 22 after infection were classified and displayed according to their related pathways (Appendix A). Those results indicated that, after activation of the MAPK signaling pathway (Figure 8), antitoxins were produced and genes related to ethylene synthesis were upregulated, thereby enhancing the defense response in Zhenghong 22. The results of qRT-PCR analyses confirmed the upregulation of *ACS6*, a key gene for ethylene synthesis, consistent with the transcriptome results (Figure 7G). At the same time, activation of defense response pathogen-associated molecular pattern (PAMP)-triggered immunity (PTI) induced expression of the gene encoding the respiratory burst oxidase, as well as genes related to the production of raw materials and energy–supply pathways. The plant–pathogen interaction signaling pathway, as well as pathways regulated by salicylic acid, ethylene, ABA, and indole were activated in Zhenghong 22. These pathways do not function independently, but instead interact to promote or inhibit each other. The shikimic acid pathway produces the main precursors for amino acids and secondary metabolites and showed a 10 times upward trend in Zhenghong 22. Amino acid and secondary metabolite synthesis pathways were activated in Zhenghong 22. This cultivar also accumulated phenylalanine, which is mainly involved in the benzoin synthesis pathway that generates flavonoids and lignin. Coniferyl alcohol is an intermediate in this pathway, and it accumulated in Zhenghong 22. Genes involved in the antibiotic synthesis pathway that generates podophyllotoxin showed significantly increased transcript levels in Zhenghong 22. We will further analyze this pathway and its metabolites in future research.

The metabolite tyramine also accumulated during the response to SN infection in Zhenghong 22. The terpenoid backbone formation pathway regulates the biosynthesis of carotenoids and promotes the synthesis of ABA, which is involved in stress adaptation. The zeatin biosynthesis pathway promotes cell division, which may be related to disease resistance, and the terpenoid and steroid biosynthesis pathway generates indole, which may initiate the defense response. Zhenghong 22 showed increased contents of monoterpenes and diterpenes after SN infection, and an increase in brassinosteroid, which promotes camalexin biosynthesis.

## 3. Discussion

After infection, the contents of various hormones increased and remained at a high level in the resistant cultivar Zhenghong 22. A sharp increase in the content of the ethylene precursor ACC was accompanied by increased transcript levels of ethylene synthesis genes and enhanced activity of ethylene synthesis pathways. Although SA plays an important role in the resistance response, there was little difference in SA content between the two cultivars. However, the content of the SA precursor trans-cinnamic acid was higher in Zhenghong 22 than in Longshu 9, and the transcript levels of defense-related genes initiated by SA were higher in Zhenghong 22. The ABA content was higher in Zhenghong 22 than in Longshu 9 both before and after SN infection. Therefore, we speculate that the ethylene/SA pathway is very important in SN resistance. The MAPK cascade signaling pathway is the most important pathway in the resistance response to SN. On the basis of our results, we speculate that secondary metabolites such as antitoxins, alkaloids, ketones, and terpenes may be toxic to SN, or participate in tropism and growth inhibition to inhibit the infection and spread of SN.

### 3.1. MAPK Cascade Signaling Pathway

The MAPK cascade signaling pathway is widely involved in the immune response of plants. It is a highly conserved signal module located downstream of eukaryotic receptors that respond to external stress [14,15,16]. Arabidopsis receptor for activated C kinase 1 (encoded by *RACK1*) functions as a MAPK scaffold protein in the absence of pathogen infection [17]. In cotton, GhMORG1 functions as a scaffold protein and interacts with *GhMKK6* and *GhMPK4* to mediate resistance to Fusarium wilt [18].

MAPKs are Ser/Thr protein kinases that participate in responses to biotic and abiotic stresses, growth and development, and signal transduction [19]. The candidate genes identified in this study included *itf14g18430.t1*, encoding a Ser/Thr protein kinase, and *itf12g18840.t1,* encoding a gene related to ethylene synthesis. Future research will focus on how the ethylene signaling pathway is regulated during SN resistance.

### 3.2. Effects of Ethylene on Plant Disease Resistance

Ethylene synthesis is regulated by the activity of ACC synthases (ACS). The conversion of S-adenosylmethionine (SAM) to ACC by ACS is considered to be the rate-limiting step in ethylene biosynthesis [20,21]. Our results show that *ACS6* was upregulated in Zhenghong 22, as was ACC. These findings suggest that ethylene production increased to regulate resistance to SN.

Jasmonic acid (JA), SA, and ethylene are key components of responses to biotic stresses and play important roles in plant disease resistance [22]. Many studies have shown that ethylene positively regulates resistance to some pathogens in plants. For example, ethylene biosynthesis was found to increase in rice plants infected with the blast fungus *Magnaporthe oryzae*, and the ethylene signal transduction element encoded using *OsEIN3* may play an important role in the production of reactive oxygen species, JA, and phytoalexins [23]. Overexpression of *TuACO3* promoted the biosynthesis of ethylene in plants and enhanced their resistance to *Blumeria graminisf*. sp. Tritici [24].

### 3.3. Synergistic Effects of Ethylene and Other Hormones on Plant Disease Resistance

Ethylene, SA, and JA are key regulators of the plant immune response. The synergy or antagonism among hormones and the intersection of related signaling pathways are current research hotspots. We found that the SA content increased rapidly in both cultivars after infection, and the ethylene biosynthetic pathway was initiated in the resistant cultivar Zhenghong 22.

It is known that SA can inhibit the conversion of ACC to ethylene [25]. A study on tomato showed that coordination between ethylene and SA caused susceptibility to pathogens [26]. Studies have shown that PTI and effector-triggered immunity (ETI) induced by pathogen infection can induce ethylene biosynthesis, and this can be enhanced using SA treatment. Both ethylene and SA were found to be involved in the immune response triggered by *Pseudomonas syringae* in *Arabidopsis thaliana* [27]. We speculate that stem nematode infection may trigger PTI-mediated ethylene synthesis and SA synthesis in sweet potato.

Ethylene and JA function synergistically to regulate plant immune responses. Apetala 2/ethylene response factor (AP2/ERF) ORA59, an important component of the ethylene signaling pathway, plays an important role in this process. Hydroxycinnamic acid amides are a class of antibacterial metabolites that are involved in the defense response of plants against necrotic pathogens including *Botrytis cinerea* and *Alternaria brassicicola* [28]. Another study showed that overexpression of *ORA59* in *Arabidopsis* increased its resistance to *B. cinerea.* ORA59 regulates the expression of JA/ethylene-related genes by directly binding to elements in the promoter of target genes to regulate plant defense against necrotic pathogens in a synergistic manner [29]. In our study, the abundance of SA precursor substances and transcript levels of SA biosynthetic genes were higher in Zhenghong 22 than in Longshu 9. Interestingly, the SA content was higher in Zhenghong 22 than in Longshu at 30 dpi, but not at 10 dpi. We speculated that the combined regulation of the two hormones led to greater consumption of SA in the resistant cultivar Zhenghong 22 during its resistance response at the early stage. In further research, we will explore the joint regulatory pathway of ethylene and SA. Previous studies have shown that ABA, brassinosteroid, and other hormones regulate plant immunity by interacting with transcription factors [30].

### 3.4. Role of Resistance-Related Metabolites

The first level of the immune response is PTI. During this process, pattern recognition receptors recognize conserved molecular patterns, such as lipopolysaccharides, peptidoglycans, chitins, flagellins, and ergosterol, collectively known as pathogen-associated molecular patterns [31,32]. They can initiate and continuously activate the innate immune response [33]. The candidate gene *itf10g03250.t1* putatively encodes a subilase family protein. Its expression pattern suggests that it is an innate immune protein.

Plants produce resistance (R) proteins, many of which are intracellular NB-LRR proteins [34]. Both pattern recognition receptors and NLR initiate downstream signaling networks, leading to the expression of defense-related genes, the production of reactive oxygen species, and callose deposition [35,36,37]. The candidate genes *itf14g17940.t1* and *itf06g12490.t1* encode LRR receptors, and *itf04g08530.t1* encodes an R protein.

In our study, Zhenghong 22 accumulated shikimic acid, which is produced by the shikimic acid pathway from phosphoenolpyruvate and erythrose-4-phosphate. Phenylalanine, tyrosine, and tryptophan are produced in the shikimic acid pathway, and are the cornerstones of protein synthesis. They are also common precursors of plant secondary metabolites such as phenolic compounds and nitrogen-containing compounds. Phenylalanine is a common precursor of flavonoids, lignans, lignin, condensed tannins, and phenylpropionic acid benzene volatiles. Tyrosine is also used in the synthesis of isoquinoline alkaloids, pigment betaines, and quinones. Tryptophan is a precursor of alkaloids, plant antitoxins, indole glucosinolates, and auxin [38].

Secondary metabolites are mainly involved in plant defense and environmental communication. In transgenic rice plants, the accumulation of phenylpropanoid and flavonoids increased or decreased to varying degrees after rice blast infection [39]. In our study, Zhenghong 22 accumulated coniferyl alcohol, an intermediate in the phenylpropanoid pathway, to high levels. Coniferyl alcohol is a precursor for podophyllotoxin biosynthesis. Although the transcript levels of two podophyllotoxin biosynthesis genes were significantly increased in Zhenghong 22, the corresponding metabolite was not detected. However, our results confirm that these substances exist and play a role. The final products of the phenylalanine metabolic branch such as flavonoids, hydroxycinnamic acid lipids, hydroxycinnamic acid amides, and precursors of lignin, lignans, and tannins play important roles in the SN resistance of sweet potato [40].

After infection, many defensive secondary metabolites are secreted by plants [37]. In this study, we detected increased contents of nicotine, monoterpenes, and diterpenes in the SN-resistant cultivar Zhenghong 22, suggesting that these compounds play an important role in SN resistance. The specific metabolites and how they affect the disease resistance of sweet potato will be explored in further studies.

## 4. Materials and Methods

### 4.1. Plant Materials

Two sweet potato cultivars were selected for this study: the resistance to both intrusive and expansive of stem nematode cultivar Zhenghong 22 and the susceptible cultivar Longshu 9. Thirty mid-sized (200–300 g) storage roots that were complete and healthy were selected for the infection experiment.

### 4.2. Stem Nematodes Infection

Diseased sweet potato storage roots were collected, and nematodes were isolated using the Bayman funnel method. The cultured nematodes were filtered using a 400-mesh sieve, washed three times with sterile water, and placed in a preprepared solution of 1% *w*/*v* penicillin and 1% *w*/*v* streptomycin. The nematodes were stored in this solution at 4 °C for 12 h, washed three times with sterile water, and then used to prepare a suspension of 200 nematodes per mL for use.

A cylinder-shaped piece (5 mm diameter) was removed from the middle of each storage root, and 200 nematodes were placed inside the hole with a sterilized pipette. After removing 1.5 cm from the bottom of the cylinder, it was replaced in the hole. The cut part was sealed with paraffin, and then the storage roots were placed in a sterilized plastic box and kept at 26–28 °C and 80–90% humidity [41].

### 4.3. Sampling and Phenotypic Observations

Samples were collected at 0 h, 0.5 d, 1 d, 3 d, 10 d, and 30 d post-infection (dpi) with stem nematodes for transcriptome analysis. The phenotype of infected storage roots was recorded at 10 dpi and 30 dpi. Samples were collected at 0 dpi, 10 dpi, and 30 dpi for metabolomic analysis.

### 4.4. Transcriptome Sequencing and Raw Data Quality Control

Total RNA was extracted using Trizol reagent (Invitrogen, Carlsbad, CA, USA) according to the manufacturer’s protocol. The quality of the extracted RNA was assessed using an Agilent 2100 Bioanalyzer (Agilent Technologies, Palo Alto, CA, USA) and RNase-free agarose gel electrophoresis. After total RNA was extracted, eukaryotic mRNA was enriched using oligo (dT) beads. The enriched mRNA was cut into short fragments using fragmentation buffer and reverse-transcribed into cDNA with random primers. Second-strand cDNA was synthesized in a mixture of DNA polymerase I, RNase H, dNTP, and buffer. The cDNA fragments were purified using a QiaQuick PCR extraction kit (Qiagen, Venlo, The Netherlands) and end-repaired before addition of poly (A) and ligation to Illumina sequencing adapters. The ligation products were size-selected using agarose gel electrophoresis, PCR-amplified, and then sequenced on the Illumina HiSeq2500 platform by Gene Denovo Biotechnology Co. (Guangzhou, China). The clean reads were compared with the reference genome (http://sweetpotato.uga.edu/index.shtml, accessed on 31 May 2022) of sweet potato’s wild relative *Ipomoea trifida*. Differentially expressed genes (DEGs) between samples were detected using DESeq2 software, with the following criteria: false discovery rate of <0.05 and absolute fold change of ≥2.

### 4.5. Metabolite Extraction, Data Processing, and Metabolite Identification

Each storage root sample (100 mg) was ground in liquid nitrogen and the homogenate was resuspended in 500 μL prechilled 80% methanol using vigorous vortexing. After processing, the supernatant was analyzed using liquid chromatography tandem mass spectrometry (LC–MS/MS) [42,43].

The LC–MS/MS analyses were performed using a Vanquish ultrahigh-performance liquid chromatograph (UHPLC) (Thermo Fisher Scientific, Waltham, MA, USA) coupled with an Orbitrap Q Exactive TM HF-X mass spectrometer (Thermo Fisher Scientific) by Gene Denovo Co., Ltd. (Guangzhou, China) [44]. The raw data files generated using UHPLC–MS/MS were processed using Compound Discoverer 3.1 (CD3.1, Thermo Fisher Scientific) to perform peak alignment, peak-picking, and quantification for each metabolite. The normalized data were used to predict the molecular formula based on additive ions, molecular ion peaks, and fragment ions. Peaks were matched with data in online spectral libraries (mz Cloud (https://www.mzcloud.org/, accessed on 31 May 2022); mz Vault) and the Mass List database to obtain accurate qualitative and relative quantitative results. Statistical analyses were performed using the statistical software R (R version R-3.4.3), Python (Python 2.7.6 version), and CentOS (CentOS release 6.6). When data were not normally distributed, the area normalization method was applied.

### 4.6. Real-Time Quantitative PCR

Candidate genes were selected for real-time quantitative PCR analyses, which were conducted on a CFX Connect instrument (Bio-Rad, Hercules, CA, USA) with HiScript II reverse transcriptase according to the manufacturer’s protocols (Vazyme Biotech Co., Ltd., Nanjing, China). The specific primers for glycosyltransferase genes were designed using Primer Premier 5.0 (Appendix A). A constitutively expressed gene (*Actin*) was used as the internal control. The relative transcript levels of genes were calculated using the 2^−ΔΔCt^ method. Three biological replicates and three technical replicates were analyzed.

### 4.7. Weighted Gene Co-Expression Network Analysis

The co-expression network was constructed using the WGCNA (v1.47) package in R. After filtering genes, we calculated the correlation coefficients between module eigengene values and samples or sample traits to identify biologically significant modules. The intramodular connection strength, module membership (MM) value, and gene significance values were calculated for each gene. Genes with high connectivity tended to be hub genes that may have important functions. Correlation analysis was performed using module eigengene values and data for specific traits or phenotypes. Pearson’s correlation coefficients between each gene and trait data were calculated for the most relevant module (positive and negative correlations) corresponding to each phenotype. The network was visualized using Cytoscape_3.3.0.

## 5. Conclusions

In this study, Zhenghong 22, a sweet potato cultivar highly resistant to SN, and Longshu 9, a highly susceptible cultivar, were analyzed to reveal the regulatory pathways involved in SN resistance using transcriptome and metabolome correlation analysis. In total, 45 key candidate genes and 32 key candidate metabolites were obtained. In the early stage (0–1 dpi), activation of the MAPK signaling pathway led to the upregulation of genes involved in ethylene synthesis and disease resistance in Zhenghong 22. The SA regulatory pathway was activated, possibly as a result of two interacting hormones. Respiratory burst oxidase activated the hypersensitive response in the resistant cultivar Zhenghong 22, and the glycolytic pathway was upregulated to provide amino acids and energy. In the middle (3–10 dpi) stage, brassinosteroid mediated camalexin production, and zeatin also played an important role. *Pti1* may have enhanced the accumulation of reactive oxygen species. At the late stage (30 dpi), the respiratory burst oxidase gene and *Pti1* were again activated. The secondary metabolite biosynthesis pathways remained active and generated various hormones and phenylpropanoids that were incorporated into lignin to participate in disease resistance, as well as other secondary metabolites such as alkaloids, monobactam, and ketones. Alkaloids, monoterpenoids, flavonoids, and other terpenoids were implicated in resistance to SN. Our results indicate that *itf12g18840,* which encodes a gene related to ACC synthesis, and *itf06g12490*, which encodes an LRR-like protein, play important roles in resistance to SN in sweet potato. Gene function identification will be the next research direction for our group.

## Figures and Tables

**Figure 1 plants-12-02052-f001:**
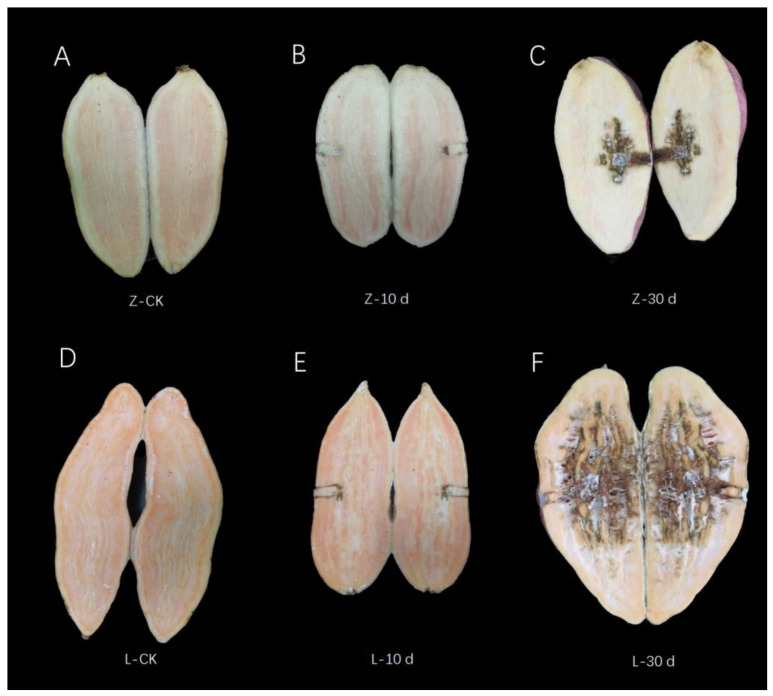
Phenotypic changes in the two cultivars after infection with stem nematode. (**A**) Phenotype of control samples of resistant variety Zhenghong 22. (**B**) Phenotype at 10 dpi of Zhenghong 22. (**C**) Phenotype at 30 dpi of Zhenghong 22. (**D**) Phenotype of control samples of sensitive variety Longshu 9. (**E**) Phenotype at 10 dpi of Longshu 9. (**F**) Phenotype at 30 dpi of Longshu 9.

**Figure 2 plants-12-02052-f002:**
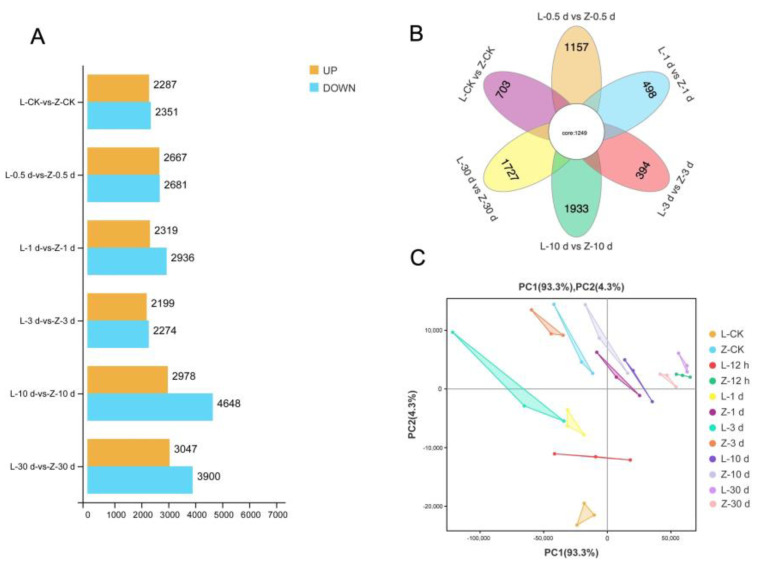
Transcriptome data analysis. (**A**) Histogram showing number of differentially expressed genes (DEGs) between two sweet potato varieties (Z: resistant; L: susceptible) after infection with stem nematodes at different times. (**B**) Venn diagram analysis of DEGs in two sweet potato varieties at different times after infection. (**C**) Principal component analysis based on samples of two cultivars (Z, L) at different times after infection.

**Figure 3 plants-12-02052-f003:**
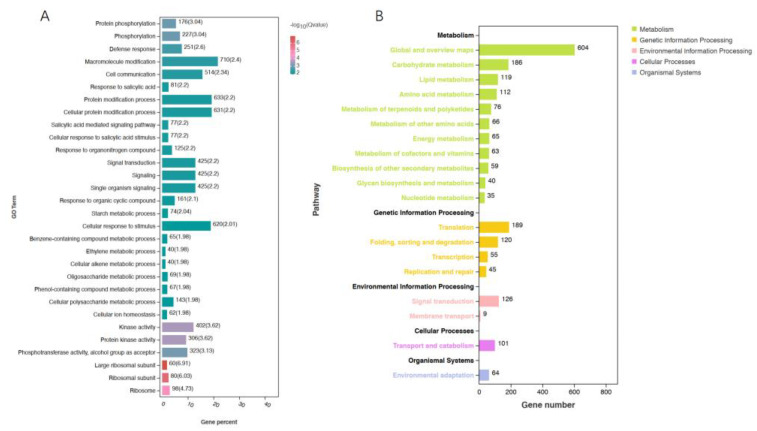
GO and KEGG enrichment analyses of DEGs. (**A**) GO enrichment analysis of DEGs. (**B**) KEGG enrichment analysis of DEGs, showing pathways most enriched with DEGs between SN-resistant and sensitive sweet potato varieties.

**Figure 4 plants-12-02052-f004:**
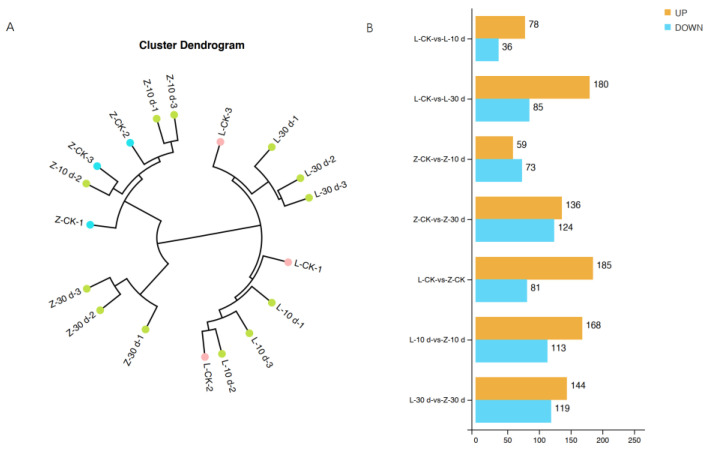
Cluster analysis of samples and number of differentially accumulated metabolites (DMs) in different comparison groups. (**A**) Cluster dendrogram of samples; (**B**) number of up- and downregulated DMs in different comparison groups.

**Figure 5 plants-12-02052-f005:**
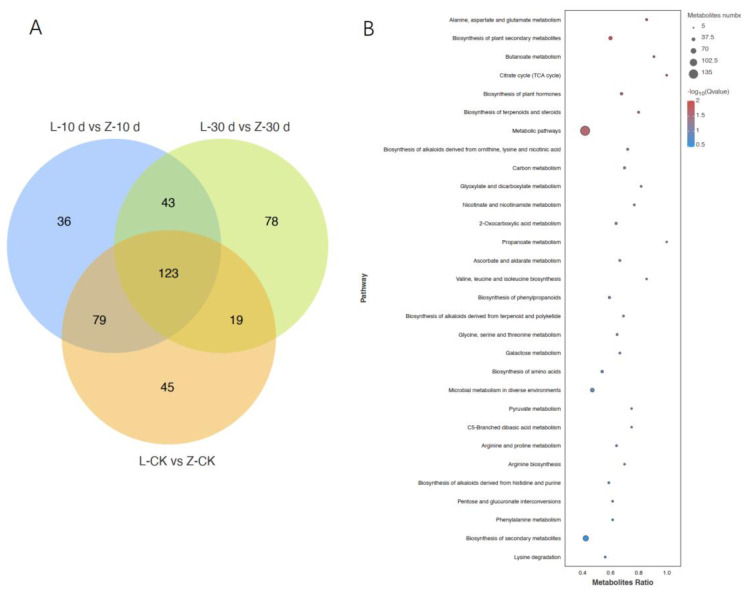
Differentially accumulated metabolites (DMs) screening and KEGG analysis. (**A**) Venn diagram analysis of DMs between resistant and sensitive varieties at different times after infection. (**B**) KEGG enrichment analysis of DMs between resistant and sensitive varieties at different times after infection.

**Figure 6 plants-12-02052-f006:**
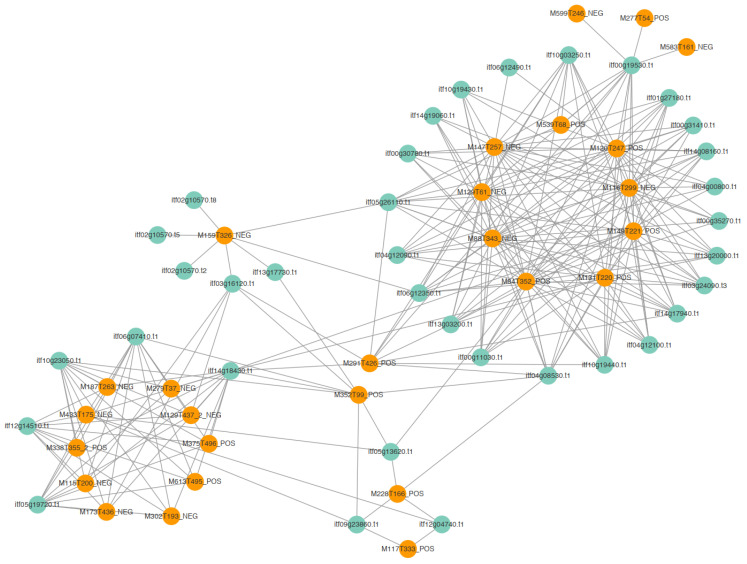
Connectivity network among candidate genes and metabolites.

**Figure 7 plants-12-02052-f007:**
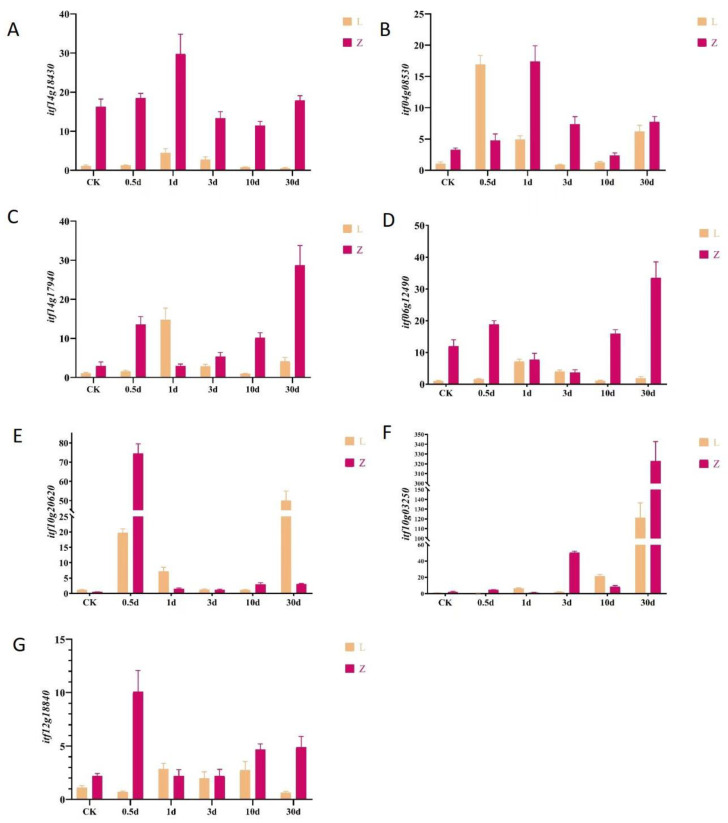
Transcript levels of genes in two sweet potato cultivars at various time points after infection by stem nematode, as determined using quantitative real-time PCR. (**A**–**F**) Transcript levels of various candidate genes selected from WGCNA. (**G**) Transcript levels of *ACS6*.

**Figure 8 plants-12-02052-f008:**
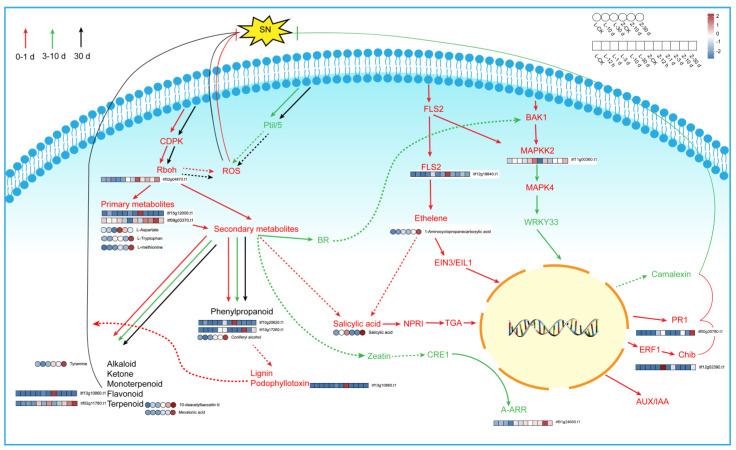
Pathway analysis of resistance to stem nematode in sweet potato and heatmap of related genes and metabolites.

**Table 1 plants-12-02052-t001:** Genes with high connectivity selected from WGCNA.

Gene	Function Annotation	Gene	Function Annotation
*itf03g17570.t1*	Myosin family protein with Dill domain	*itf05g19720.t1*	Homeobox
*itf13g17730.t1*	Ubiquitin protein ligase	*itf10g23050.t1*	Inositol monophosphatase family protein
*itf05g01680.t1*	RNA polymerase	*itf12g14510.t1*	Hypothetical protein
*itf04g27140.t1*	Cytochrome P450, family 82	*itf13g03200.t1*	Osmotin
*itf07g02790.t1*	HXXXD-type acyl-transferase family protein	*itf13g20000.t1*	Eukaryotic asparty-l protease family protein
*itf03g00310.t1*	Lysine histidine transporter	*itf14g17940.t1*	LRR-transmembrane protein kinase
*itf15g12000.t1*	Long-chain acyl-CoA syntheses	*itf04g12100.t1*	Glycosyl hydrolase superfamily protein
*itf15g12000.t2*	Long-chain acyl-CoA syntheses	*itf04g12090.t1*	Glycosyl hydrolase superfamily protein
*itf10g20620.t1*	Peroxidase super family protein	*itf00g11030.t1*	Disease resistance protein family
*itf01g06350.t1*	LRR-transmembrane protein kinase	*itf00g35270.t1*	Unknown
*itf04g08530.t1*	MLP-like protein	*itf04g00800.t1*	Chitinase A
*it06g24000.t1*	Oligopeptide transporter	*itf06g12490.t1*	LRR-transmembrane protein kinase
*itf12g04740.t1*	Cyclin D3; 3	*itf10g03250.t1*	Subtilase family protein
*itf05g13620.t1*	Lipid transporters	*itf10g03260.t1*	Subtilase family protein
*itf09g23860.t1*	Aldolase-type TIM barrel family protein	*itf00g35270.t1*	Unknown
*itf04g13120.t1*	Bidirectional amino acid transporter	*itf00g31410.t1*	Cysteine-rich PLK
*itf00g32090.t2*	Elicitor-activated gene 3-2	*itf10g19440.t1*	Senescence-associated gene
*itf14g18430.t1*	NIMA-related kinase	*itf10g19430.t1*	Cysteine proteinases superfamily protein
*itf06g07410.t1*	Unknown	*itf14g08160.t1*	Receptor-like protein kinase

**Table 2 plants-12-02052-t002:** Candidate metabolites selected by different methods.

WGCNA	Pink from the Trends	Pink from 10 Times Difference
1-octadecanoyl-sn-glycero-3-phospho-(1’-myo-inositol)	cis-9-Palmitoleic acid	Shikimate
2,5,7,8-tetramethyl-2-(beta-carboxyethyl)-6-hydroxychroman	Coniferyl alcohol	L-Tryptophan
9s-hydroperoxy-10e,12z-octadecadienoic acid	Cucurbitacin d	Cis-aconitate
Deoxycytidine	Dimethylglycine	Malonic acid
Desferrioxamine d2	trans-cinnamate	Perillyl alcohol
Gambogic acid	Tyramine	Coniferyl alcohol
Glu-Ala-Arg	Indole	5,6,7,8-tetrahydro-2-Naphthoic Acid
Naringenin-7-o-glucoside	Ketoleucine	13-keto-9z,11e-octadecadienoic acid
Succinamide	Mevalonic acid	Prostaglandin i2
Trans-zeatin-riboside	Sarcosine	5a,6-anhydrotetracycline
Tyr-Arg	1-Aminocyclopropanecarboxylic acid	Phosphorylcholine
Val-Ala	Argininosuccinic acid	
Val-Trp	2-oxoadipic acid	
	Glutathione, oxidized	
	N-alpha-acetyl-l-ornithine	
	Cis-aconitate	
	Citraconic acid	
	Fumarate	
	Linoleic acid	

## Data Availability

The data presented in this study are available on request from the corresponding author.

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
