# Peer review of "Integrated Transcriptome and Metabolome Analyses Reveal Details of the Molecular Regulation of Resistance to Stem Nematode in Sweet Potato"

_plants, 2023, doi:10.3390/plants12102052_

Round 1

Reviewer 1 Report

1. The reference list should adjust and rearrange it in the correct order.

for example, Line 51: ‘stress[3,7,8]. Qu et al[4]

2. Please pay attention to writing standards and add a space between numbers and units, for example, Line 84, in figure 1, L-10d shoule be L-10 d. Please carefully check the entire text including the figure and ensure that spaces are added between numbers and units where necessary.

3. Please adjust the position of Figures 7 and 8 in the manuscript and place them in their respective sections. It is not suitable for placement on the discussion page.

The overall language and expression of the article need to be further revised and condensed.

Author Response

Point 1: The reference list should adjust and rearrange it in the correct order. for example, Line 51: ‘stress[3,7,8]. Qu et al[4]

Response 1: We have adjusted the references to the correct order.

Point 2: Please pay attention to writing standards and add a space between numbers and units, for example, Line 84, in figure 1, ‘L-10d’ shoule be ‘L-10 d’. Please carefully check the entire text including the figure and ensure that spaces are added between numbers and units where necessary.

Response 2: we have carefully checked the entire text including the figure and ensure that spaces are added between numbers and units.

Point 3: Please adjust the position of Figures 7 and 8 in the manuscript and place them in their respective sections. It is not suitable for placement on the discussion page.

Response 3: The position of Figures 7 and 8 in the manuscript has been adjusted.

Reviewer 2 Report

Overall the document is well written and structured and the ideas are generally well developed. The document may be accepted for publication after some minor corrections.

Some suggested minor revisions:

Line 42 – “(Ditylenchus destructor Thorne, SN)” – Thorne is not in italics

Line 217 - Remove “e” in “formulae”

Line 254 – Change “figure. 6” to “Fig. 6”

Line 343/344 - A blank space between line 343 and 344.

Author Response

Point 1: Some suggested minor revisions:

Line 42 – “(Ditylenchus destructor Thorne, SN)” – Thorne is not in italics

Line 217 - Remove “e” in “formulae”

Line 254 – Change “figure. 6” to “Fig. 6”

Line 343/344 - A blank space between line 343 and 344.

Response 1: All formatting errors have been corrected and the manuscript carefully examined
